# Miniaturized Salinity Gradient Energy Harvesting Devices

**DOI:** 10.3390/molecules26185469

**Published:** 2021-09-08

**Authors:** Wei-Shan Hsu, Anant Preet, Tung-Yi Lin, Tzu-En Lin

**Affiliations:** 1Institute of Biomedical Engineering, College of Electrical and Computer Engineering, National Yang Ming Chiao Tung University, Hsinchu 30010, Taiwan; a0919560481@gmail.com (W.-S.H.); or d09223205@ntu.edu.tw (A.P.); 2Chemical Biology and Molecular Biophysics Program, Taiwan International Graduate Program, Institute of Biological Chemistry, Academia Sinica, Nankang, Taipei 115, Taiwan; 3Department of Chemistry, College of Science, National Taiwan University, Taipei 10617, Taiwan; 4Institute of Traditional Medicine, College of Medicine, National Yang Ming Chiao Tung University, Taipei 11221, Taiwan; tylin99@nycu.edu.tw; 5Program in Molecular Medicine, College of Life Sciences, National Yang Ming Chiao Tung University, Taipei 11221, Taiwan; 6Biomedical Industry Ph.D. Program, College of Life Sciences, National Yang Ming Chiao Tung University, Taipei 11221, Taiwan

**Keywords:** salinity gradient energy, osmotic energy, blue energy, energy harvesting, reverse electrodialysis (RED), ion-exchange membranes, nanofluidic membranes

## Abstract

Harvesting salinity gradient energy, also known as “osmotic energy” or “blue energy”, generated from the free energy mixing of seawater and fresh river water provides a renewable and sustainable alternative for circumventing the recent upsurge in global energy consumption. The osmotic pressure resulting from mixing water streams with different salinities can be converted into electrical energy driven by a potential difference or ionic gradients. Reversed-electrodialysis (RED) has become more prominent among the conventional membrane-based separation methodologies due to its higher energy efficiency and lesser susceptibility to membrane fouling than pressure-retarded osmosis (PRO). However, the ion-exchange membranes used for RED systems often encounter limitations while adapting to a real-world system due to their limited pore sizes and internal resistance. The worldwide demand for clean energy production has reinvigorated the interest in salinity gradient energy conversion. In addition to the large energy conversion devices, the miniaturized devices used for powering a portable or wearable micro-device have attracted much attention. This review provides insights into developing miniaturized salinity gradient energy harvesting devices and recent advances in the membranes designed for optimized osmotic power extraction. Furthermore, we present various applications utilizing the salinity gradient energy conversion.

## 1. Introduction

### 1.1. The Blue Energy: Salinity Gradient Energy

The requirement for clean and sustainable energy sources to address the overconsumption of fossil fuels has been critical over recent years [1]. Considerable concerns have been raised regarding the side-effects resulting from the excessive utilization of fossil fuels leading to global warming and environmental pollution. Thus, alternative renewable and sustainable energy sources have attracted worldwide attention, and tremendous efforts have already been made to utilize “green energy” sources such as solar, wind, tidal, geothermal, ocean thermal, and biomass [2,3,4]. Salinity gradient energy (SGE), which is also known as “blue energy” or “osmotic energy”, is a type of renewable energy that is produced when two aqueous solutions with different salinities mix with each other and thereby, create an osmotic pressure that can be converted to mechanical or electrical energy [5]. For instance, power plants for collecting enormous potential difference are developed from mixing highly saline ocean water with low-saline river water (freshwater) [6]. It has been estimated that the global salinity difference between freshwater and ocean water comprises enormous osmotic energy of approximately 2 trillion watts (TW), which is equivalent to that of 2000 atomic reactors, enough to meet global energy needs [4,7,8,9]. However, further in-depth knowledge is still required to understand how to extract salinity gradient power efficiently.

Recently, SGE is becoming more popular as it is readily available (24 h a day) and feasible compared to conventional unstable energy sources such as wind, solar, and wave, which exhibit high daily variability [10]. Remarkable progress is being made to highlight the significance of blue energy harvesting and expanding its utilization in real-life applications, including energy storage and powering electrical devices [11,12].

Indeed, in addition to optimizing osmotic energy production, it is crucial to maintain a balanced aquatic ecosystem when using blue energy. To avoid the excessive utilization of aquatic resources and interference to the living creatures in the sea environment, other naturally occurring hypersaline sources (such as hypersaline lakes and salt domes) can be explored for obtaining a greater concentration difference and, hence, provide solutions to overcome the challenges inherent to seawater-river water systems [13]. 

The salinity gradient energy relies on the Gibbs free energy of mixing, which can be expressed by Equation (1) as follows: (1)$$\;\Delta_{mix}G=\Delta G_{b}\;–\;\left(\Delta G_{c}+\Delta G_{d}\right)$$
where the subscripts “**b**”, “**c**”, and “**d**” denote the saline solution resulted after mixing, concentrated solution, and dilute solution, respectively [5].

The relationship between the entropy of mixing can be expressed by Equation (2) as follows:(2)$$\;\;\Delta_{mix}G=-\left(n_{c}+n_{d}\right)T\Delta_{mix}S_{b}\;–(-n_{c}T\Delta_{mix}S_{c}-n_{d}T\Delta_{mix}GS_{d\;})$$
where **n** denotes the number of particles (mol), **T** denotes the temperature (in Kelvin), and **Δ**_**mix**_**S** denotes the molar entropy of mixing (J∙(mol∙K)^−1^) [14].

Numerous sustainable methodologies have been developed to utilize this naturally abundant energy source, out of which two membrane-based separation processes, including reverse electrodialysis (RED) and pressure-retarded osmosis (PRO), are the most frequently used for experimental studies [15]. The PRO-based system comprises semi-permeable membranes allowing only the water molecules from a lesser saline solution (e.g., river) to pass through them for generating the chemical potentials to reach equilibrium, hence generating mechanical energy from the movement of water molecules, which can later be converted into electricity [4]. However, it poses a risk of membrane fouling, leading to a high cost of operation [5]. On the other hand, the RED-based system involves the ion-exchange membranes or charged channels that allow the anions/cations to selectively pass through them to generate the electrochemical potential difference that can be converted to the flow of electrons via redox reactions or capacitive electrodes [4,5,16,17,18].

Furthermore, RED-based technologies are relatively more desirable for osmotic power generation as they offer a higher energy efficiency [7,8,19,20]. A typical RED stack designed to extract salt gradient energy comprises anion-exchange membranes (AEMs) and cation-exchange membranes (CEMs) placed in an alternating series. This arrangement forms the flow compartments of low-salinity and high-salinity streams to facilitate the ionic flux and can be connected to an external electric load (as depicted in Figure 1). These flow streams caused by ion-selective diffusion and osmotic pressure can lead to the separation of cations and anions and, thus, drive the redox reaction on the surface of the electrodes.

Recent advances in the development of ultrathin, cost-effective, and energy-efficient membranes have enabled their utilization in optimized osmotic energy conversion devices [20,21,22,23,24,25,26]. Nowadays, the applications of SGE harvesting devices have been broadened to numerous sectors beyond power plants near the coasts, for instance, biomedical applications [27]. Hence, this review systematically summarizes the modern development of miniaturized SGE harvesting devices. Herein, we emphasize the potential and future applications of RED technology.

### 1.2. The History of the Development of Salinity Gradient Energy Devices

The concept of salinity gradient power was first introduced in 1954 by Pattle [28]. He found the links between the free energy of mixing and the electric power resulting from mixing fresh and saltwater [28]. As this technology progressed, optimizing the efficiency of ion-selective membranes for collecting energy or desalination became more vital [29]. The area and quantities for fabricating commercial power plants are at least 2 million m^2^ of membrane surface for a 2 MW power plant [30,31,32]. Over the years, the replacement and maintenance of such plants over time (about 5 years) encountered various challenges, and thus, the progress of the RED technology has been made (as shown in Figure 2) [7]. How to boost the efficiency of the power plant is still a critical issue [11]. Various leading companies such as Fujifilm, FumaTech, Nitto Denko, Oasys Water, OsmoBlue, Pentair X Flow, Porifera, and Toray Industries emphasize the production of membranes with improved efficiency and economic feasibility.

In recent decades, people noticed that the nature and biological systems have proven to be our best teachers as they taught us how to work more efficiently. Ionic systems are regarded as one of the fundamental systems for governing physiological processes in all living creatures [37]. For instance, the nerve signals are regulated by ion channels that transport cations or anions such as potassium ions, sodium ions, and chloride ions [37]. The salt gradients of these ions are strictly controlled, and the electrical signals play an essential role in aiding the communication among the nerve cells and muscles [37]. Inspired by the nerve cells, researchers decreased the sizes of salt gradient harvesting devices, and thus, numerous studies have been reported that aim to advance such systems [38,39,40,41,42]. For instance, biomimetic ion-selective membranes were synthesized, and the versatile microfluidics channels for ion exchange were designed [43]. Compared with the traditional power plants, these devices are usually more lightweight, environmentally friendly, and biocompatible.

Furthermore, other energy harvesting technologies related to the salt gradient include diffusiophoresis and diffusioosmosis, governed by electrokinetic phenomena. They have been used in numerous applications, including biomolecule separation, water purification, surface coating, and particle delivery. Diffusiophoresis refers to the spontaneous movement of colloidal particles or molecules generated due to the concentration gradient of another substance. Diffusioosmosis refers to the flow of fluids for a fixed wall or pore surface (i.e., solid/fluid interface) driven by a concentration gradient in the solution. The diffusio-osmotic flux generated from the salt concentration difference can develop osmotic currents in the tubes (BNNT) [44]. Ajdari et al. investigated the neutral solute diffusioosmosis in a slab geometry considered for non-trivial couplings between the interfacial structure and hydrodynamic slip, resulting in the faster transport of solute particles in an externally applied or self-induced concentration gradient and enhanced flow in microfluidics [45]. Additionally, diffusioosmosis circumvents the challenges faced while using sensitive electronic devices in conventional zeta potential measurements [46]. Hence, it has been shown that the salinity gradient also drives diffusion-osmotic slip flow, possibly increasing the energy conversion efficiency [45,47].

The large-sized salt gradient energy harvesting devices have already been discussed for decades. Therefore, in this review, we summarized the recent development of miniaturized salinity gradient harvesting devices. We provide an overview of how these easily activated devices can generate electricity with fascinating patterns of salt gradients. Additionally, we emphasized the innovations that have been made to optimize the ways of extracting clean and renewable energy using ion-exchange membranes and circumventing the conventional challenges, including their poor structural stability, high internal resistance, low efficiency, low ionic selectivity, and low output voltage. Hence, this review provides insights into the development of smart membrane-based salinity gradient energy harvesting devices.

## 2. Miniaturized SGE Devices

### 2.1. Membrane-Based Salinity Gradient Energy Devices

Membrane-based devices are widely used for large-scale as well as miniaturized energy harvesting instruments [3]. Typically, the mixing of salty water and freshwater can be controlled by placing the selectively permeable membranes between the salty and freshwater chambers. If the two aqueous solutions with different salt concentrations are mixed, the sum of the chemical potentials of the original, i.e., unmixed solutions minus the chemical potential of the final mixture, is the difference in free energy upon mixing [3]. Taking NaCl as an example, the free energy of mixing under constant pressure and temperature can be expressed by Equation (3) as follows:(3)$$\Delta E_{mix}=2R_{g}T\left(c_{c}ln\frac{c_{c}\left(1+\varphi \right)}{c_{c}+\varphi c_{d}}+\varphi c_{d}ln\frac{c_{d}\left(1+\varphi \right)}{c_{C}+\varphi c_{d}}\right)$$
where *R_g_* is the universal gas constant; *T* is the absolute temperature; *c_d_* and *c_c_* are the solute concentrations in the dilute and concentrated solutions, respectively; and $\varphi =\frac{V_{d}}{V_{c}}$ is the dilution ratio, with *V* the volume of the solutions being mixed [3].

According to the equation, ~0.5 kWh of energy can be generated by mixing equal volumes (1 m^3^) of seawater (~0.5 M) and fresh river water (~5 mM) [3]. This energy is equivalent to 1 m^3^ of water flowing down a 175-meter waterfall [3]. However, in a limited space and with a limited volume of liquid, the design of the membrane plays a critical role. Usually, the ion selectivity depends on the chemical functional group on the surface of the porous membranes [48]. The functional groups carry positive or negative charges, which repel the ions with the same charge but allow the ions with opposite charges to pass through the membrane. Table 1 summarizes the types and functional groups of the ion-selective membrane. The surface can be charged with positively charged functional groups or surfaces (such as positively charged aligned cellulose nanofibers, MXene surfaces, hybrid Janus nanochannel membranes). On the other hand, cation-selective membranes can be obtained via functionalizing the surfaces with negatively charged groups (such as -OH, -F, -COOH, -SO_3_^−^).

As listed in Table 1, various membranes have already been successfully developed and utilized in the salt gradient harvesting devices. However, some of those materials are either not environmentally friendly or not biocompatible. Therefore, in the following paragraphs, we describe the examples of miniaturized energy harvesting devices consisting of different biocompatible membranes.

#### 2.1.1. Salinity Power Generation Using Biocompatible Asymmetric Polypyrrole Membrane

To address the challenges faced due to the membrane fouling in PRO and RED-based systems, membranes developed from conducting polymers, such as polypyrrole (PPy), display a promising potential as they exhibit excellent electrical conductivity environmental stability, and a simple synthesis process. Composite membranes based on PPy serve as remarkable selective ion-exchange membranes [64,65]. Previous studies have reported that hybrid membranes with asymmetric structures (such as channel sizes and surface charge distribution) resemble the nanosized asymmetric ion channels in living organisms, and hence, exhibit enhanced functionality as well as a decreased ion-polarization effect [11,29,66,67,68,69,70,71,72]. Interestingly, Yu et al. developed a unique system for salinity power generation by utilizing a one-component porous PPy membrane with an asymmetric structure that displayed high biocompatibility and electroactivity [73].

The harvested salinity power being utilized onto the electric load can be measured by Equation (4) as follows:(4)$$P_{out}=I^{2}\times R$$
where *I* is the current flowing through the resistor.

For instance, the power density and current density that resulted from the salinity gradient with a PPy membrane by mixing artificial ocean water (0.5 M NaCl) and freshwater (0.01 M NaCl) attained a peak value of 0.087 W/m^2^ and 2 A/m^2^, respectively, in the presence of an outer resistance of ~5 kΩ (as shown in Figure 3) [73]. Thus, the PPy membranes exhibited excellent potential in harnessing the salinity power owing to their high ion-selectivity and desired ion transportation. Although the harvested power density using a PPy membrane is relatively lower than those calculated for heterogeneous membranes (carbon/alumina, graphene oxide, engineered polymer polyphenylsulfone), it is still higher than the reported green energy harvested via a graphene/humic acid cell [73]. Hence, a single component PPy membrane can develop smart membrane-based osmotic energy harvesting devices that can mimic biological ion transport.

#### 2.1.2. Osmotic Power Generation with Ionized Wood Membranes as Micro- or Nanofluidic Membranes

Nanofluidic channels, i.e., the ion-exchange membrane with charged nanochannels, serve as an excellent platform for salinity gradient power generation as they facilitate the fluidic transport characteristics at a nano-scale [59] To circumvent the high-cost and scalability issues faced with conventional membranes, ionized wood membranes have been explored as ion-exchange membranes for systematic osmotic energy conversion since they unveil a unique ion-transport comportment that has caught the attention of many researchers worldwide [2]. The wood membrane keeps up the well-aligned nanochannels of the cellulose nanofibers obtained from the natural wood (as shown in Figure 4). The exterior of the nanochannels can be modified to carry positive or negative charges by in situ regulation of the hydroxyl chains on the cellulose groups to quaternary ammonium or carboxyl chains, respectively [8]. Thus, efficient nanofluidic passages can be created for preferred ion flux across the membrane, resulting in the desired electrochemical potential gradient. Such nanofluidic channels comprise a minimum of 1D (<100 nm) and a double layer (DL) of ions developed at the solid–liquid interface after exposure to a fluid to achieve optimal ionic conductance during RED power generation. In situ surface functionalization can transform the natural wood membranes with macro-/meso-structures into positively or negatively charged ion-exchange membranes. The ionized wood membranes offer tunable ion-selectivity and remarkable mechanical properties, making fabrication easier and scalable for large-scale energy conversion. Furthermore, nanofluidic membranes have been explored for tackling membrane fouling and low-energy output in membrane-based processes (PRO and RED) as they can offer superior anti-fouling and antimicrobial properties [2,10].

#### 2.1.3. Enhanced Osmotic Energy Harvesting Using 2D-Composites as Nanofluidic Channels

MXenes, 2-Dimensional transition-metal carbides or nitrides, have also been investigated for their utilization in osmotic energy conversion as they display excellent physicochemical behavior [74,75,76,77]. MXene nanosheets have abundant surface groups (such as −F, −OH, and −O) that exhibit outstanding water dispersity and high surface charge density [77,78,79,80]. To address the challenges of the low stability, high internal resistance, and low selectivity of conventional membranes, Yang et al. utilized a Ti_3_C_2_T_x_ MXene/boron nitride (MXBN) membrane for boosting the salinity gradient energy conversion as the addition of BN into a pristine MXene membrane remarkably reduces the internal resistance and thereby, significantly increases the output power density [53]. Furthermore, Ti_3_C_2_T_x_ MXene membranes have been shown to harness the osmotic energy effectively as their subnanometer ion-selective channels facilitate the preferential cation exchange regulated with the modified surface terminal groups controlled via the salinity gradient. Hong et al. demonstrated a high-performance osmotic power (~21 W·m^–2^) generation governed by functionalized surface charges at a 1000-fold salinity gradient [81]. In addition, 2D Ti_3_C_2_T_x_ MXene membranes display flexibility, excellent mechanical strength, hydrophilic surfaces, and optimum electrical conductivity, making them suitable for membrane-based separation processes [10,80]. Ding et al. demonstrated that oppositely charged Ti_3_C_2_T_x_ MXene membranes with 2D nanofluidic ion-channels could be arranged in an electrochemical cell to harness the osmotic energy resulting from ionic solutions with different salinity gradients, such as seawater and river water. The negatively charged pristine MXene and positively charged MXene with surface modification act as excellent cation-selective and anion-selective membranes, respectively, facilitate the ionic transport governed by their surface charges (as shown in Figure 5) [51].

Moreover, Ji et al. developed a 2D-material-based nanofluidic RED system that comprises positively or negatively charged ion channels embedded into layered graphene oxide membranes (GOMs) for demonstrating osmotic energy conversion [52]. Moreover, recently, Zhang et al. demonstrated the utilization of black phosphorus (BP) membranes for osmotic energy conversion as the oxidation of BP promotes osmotic power generation [82]. 2D-layered BP has been explored for developing 2D nanofluidic ion transporting membranes owing to its excellent flexibility, electrical conductivity, large surface area, hydrophilic nature, and net charge delivered by the oxidized states enhancing the ionic separation in nanofluidic channels [82,83]. Hence, bioinspired 2D nanofluidic systems offer inefficient energy conversion, water treatment, and various biomedical applications [52,57,60,84,85,86,87].

### 2.2. Colloid-Based Salinity Gradient Energy Devices

Bioelectricity has always been fascinating and inspiring, and electrophysiology has progressed rapidly due to the advancements in precise and delicate experiments in various research fields [88,89]. Among the different phenomena of bioelectricity, the electric organs (EOs) of electric fishes are especially attractive because of the generation of a more powerful electric discharge. The electric eel (*Electrophorus electricus*) is a special animal species that uses EOs to generate tunable electric fields for communication, navigation, predation, and defense [49]. The EO comprises electrocytes (muscle-derived biological batteries), providing a combined voltage of up to 600 V [49,89,90]. Researchers in earlier times primarily attempted to understand the basic mechanism of bioelectricity, identify acetylcholine receptors, and eventually discuss the evolution of EOs. Although a single electrocyte of an EO can only produce a low voltage, it can add up to a high potential difference when stacked in a sequence. Hence, gaining deeper insights into the function of EOs has led to the invention of chemical batteries and the development of portable energy sources [49].

Numerous efforts have been made to establish a power-generation mechanism similar to the one adopted by electric eels. For instance, Schroeder et al. proposed a soft power source based on stacked hydrogels with different ionic gradients for chemical energy storage. Their idea was inspired by the electric eel electrophorus, a power generation system optimized by natural selection. The design of the acrylamide hydrogels, which mimic the intracellular and extracellular parts of the eel cell, is demonstrated in Figure 6. The cation and anion-selective hydrogel membranes resemble the cellular compartments with sodium and chloride selective ion channels. Ion-selective gels allow the passing of either sodium or chloride from the high salinity part of the gels to the low salinity part of the gels, resulting in a potential difference on both sides of the electrochemical cell of the power source. The principle of ion selection is that the charged functional groups on the ion-selective membrane callows the counter ions to be more permeable and can move across the membrane, while it rejecting ions with the same charge. This is know as principle of Donnan exclusion. The concentration relationship in the equilibrium state by the Donnan effect with the ion-selective membrane between I space and II space is governed by Equation (5) as follows:

(5)$${\left(\frac{C_{AI}}{C_{A\mathrm{II}}}\right)}^{1/Z_{A}}={\left(\frac{C_{BI}}{C_{B\mathrm{II}}}\right)}^{1/Z_{B}}$$
where $C_{A\;I}$ and $C_{B\;I}$ are the concentrations of “A” and “B” in the “*I”* space, $C_{A\;II}$ and $C_{B\;II}$ are the concentrations of “A” and “B” in the *II* space, and $Z_{A}$ and $Z_{B}$ are the ionic charge number [91].

Since an electrochemical cell of soft power sources produced a low voltage, 143 ± 3 mV, they could be improved further, as inspired by the electric organ of the squid to connect them into multiple sets of soft power sources in a series to achieve high potential. Figure 7 shows the electric organ of the electrophorus and the soft power sources mimicking the functionality of EOs.

Recently, Lin et al. developed a biomimetic polysaccharide battery coupled with a self-charging triboelectric nanogenerator (TENG) to demonstrate salinity gradient energy conversion into electrical energy for biomedical applications (such as disinfection). They utilized potassium chloride (KCl)-absorbed agarose hydrogels with different saline concentrations and a cation-selective gellan gum (GG) membrane for creating a salinity gradient that can induce an electrochemical potential difference (~177 mV), as shown in Figure 8 [36]. Hydrogels provide a three-dimensional (3D) network structure exhibiting the characteristics of “soft and wet” materials and hence, promote the rapid mass transfer of ions or electrons across them in the presence of a chemical gradient (such as salinity gradient) [12,54]. Moreover, Zhang et al. demonstrated using a 3D hydrogel interface for obtaining an excellent osmotic energy conversion via the hybridization of a polyelectrolyte hydrogel and aramid nanofiber membrane with enhanced interfacial transport efficiency. These heterogeneous membranes can generate high power densities (5.06 W m^–2^) when harnessing the salinity gradient energy from mixing ocean and river water [11].

### 2.3. Other Salinity Gradient Energy Devices

An alternative approach to generate electricity is by using the salinity gradient energy caused by microchannels. Recently, Feng et al. demonstrated osmotic pressure power generation using a novel material—a single-layer MoS_2_ nanoporous membrane (Single-layer MoS_2_) [86]. Nanopores (after this referred to as semi-permeable membranes) are used as a thin film in the middle of the osmotic pressure difference. This semi-permeable membrane in such a system is usually only 3 atoms thick. If this technology is used with the newly invented micro-osmotic pressure generator, it can be placed in seawater and at the mouth of the freshwater meeting. Thus, the semi-permeable membrane with an area of 1 square meter (with a nanopore coverage of 30%) can generate approximately 1MW of electricity [35]. However, to expand the application of new technologies, we first need to address the challenge of generating many relatively uniform nanopores. Wang et al. proposed a high voltage nanofluidic energy generator enlivened by the electrical eel utilizing the particle focus angles, which changes over Gibbs free energy into power without the formation of toxic substances [35].

On the other hand, the high voltage can be instigated by a restricted microscale space by multi-stacking cation and anion-trade nanochannel network layers (CE-NCNMs and AE-NCNMs). These films were built by in situ self-collected nanoparticles with hydroxyl and amine gatherings, respectively [90]. The different piles of CE-NCNMs and AE-NCNMs were effectively acknowledged by decisively managing the nano drops with the suspended decidedly or contrarily charged nanoparticles into the ideal situations in the various microchannel stage. The proposed nanofluidic energy generator presentation was quantitatively explored by changing the nanoparticle species, intermembrane distance (IMD), and natural temperature.

Furthermore, Sui et al. developed an asymmetric hybrid nanochannel membrane made from an ultrathin nanoporous polymeric membrane with carboxyl groups on the nanochannels’ surface and the amphoteric anodic alumina oxide (AAO) nanochannels with hydroxyl groups [27]. A synergistic effect obtained from such hybrid nanochannels results in diode-like ionic transport, and hence, high-power osmotic energy conversion can be facilitated in a wide pH range. For instance, at pH 3, 7, and 11, the power densities reached 2.65, 2.57, and 2.94 W/m^2^, respectively.

Hence, the design of ion-selective membranes can be improved and made scalable by mimicking the naturally occurring bioelectric systems, as in electric eels. Thereby, the asymmetric nanochannel membranes, MXene surfaces, ionized wooden membranes, and single-layer MoS_2_ nanoporous membranes can be used to obtain efficient osmotic energy conversion.

## 3. Discussion and Outlook

Membrane-based reverse electrodialysis provides a promising approach for osmotic energy conversion. The challenges encountered while using conventional membranes with a low output and high interfacial resistance can be circumvented greatly by using smart hybrid membranes that elicit long-term aqueous stability, efficient ionic transport, and high voltage output. For developing high-yield osmotic energy conversion systems, the size and shape of the channels should also be taken into account. In addition, the synergy between the geometrical parameters (such as size, shape, curvature, length) and the optimal experimental parameters (such as pH, temperature, concentration difference, salt type) can result in a better design of the salinity gradient energy devices. Furthermore, these devices can be used as biomimetic systems by utilizing membranes that exhibit excellent biocompatibility.

## 4. Conclusions

In summary, we have emphasized how salinity gradients can provide a sustainable approach for energy storage. Both theoretical and experimental studies have validated that by placing a semi-permeable membrane between two salt solutions of different concentrations, an electrochemical potential difference can be created as the low-concentration solution penetrates the high-concentration solution until the chemical equilibrium is attained. This principle can be applied to generate a large-scale osmotic pressure by the optimal mixing of ocean water and river water. Progress in the development of bio-inspired ion-selective membranes is unraveling the design of viable osmotic power generators.

## 5. Future Perspectives

While salinity gradient energy sources can provide environmentally friendly, cost-effective, and renewable energy production, efforts still need to be made for optimal large-scale production. Further improvements can still be made to optimize the osmotic energy harvesting by improving the membrane that regulates the spontaneous mixing of the aqueous saline solutions. In addition to enhancing the membrane power density, the improved material design and manufacturing methods can provide a high voltage output. Furthermore, a significant amount of osmotic energy can be harnessed from the organic solutions with inorganic salts, although their scalability issue still needs to be addressed.

## Figures and Tables

**Figure 1 molecules-26-05469-f001:**
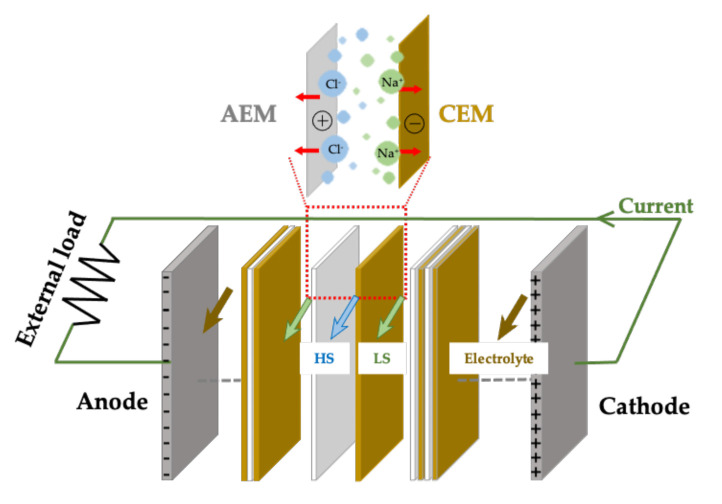
A typical RED stack being connected to an external electric load.

**Figure 2 molecules-26-05469-f002:**
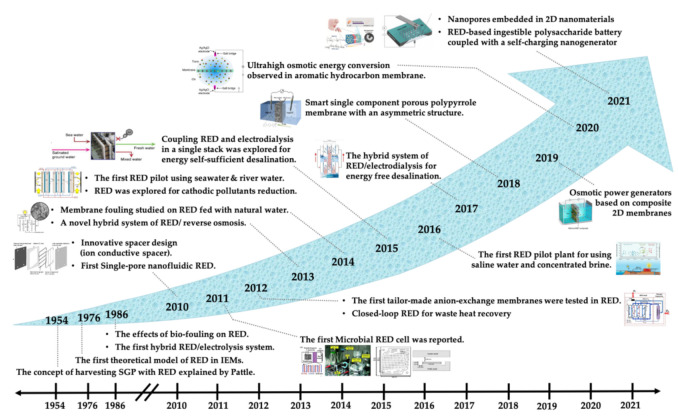
The history of RED-based salt gradient energy applications [7,33,34,35,36] Reprinted from ref. [7]. Copyright 2018, Elsevier; permission conveyed through Copyright Clearance Center, Inc.

**Figure 3 molecules-26-05469-f003:**
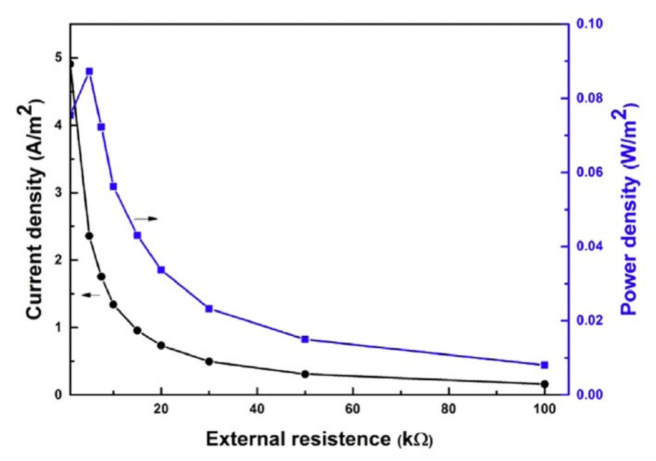
The harvested salinity power was obtained by mixing river water and artificial ocean water with a 0.01 M/0.5 M concentration gradient [73]. Reprinted from ref. [73]. Copyright 2018, Elsevier; permission conveyed through Copyright Clearance Center, Inc.

**Figure 4 molecules-26-05469-f004:**
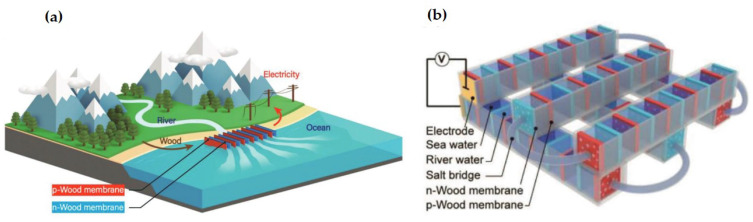
(**a**) Schematic illustration of the wood-based RED system. The wooden membranes were sequentially arranged, with salt bridges linking the adjacent wooden membrane assemblies. (**b**) The illustration of the large wood-based RED system for osmotic energy conversion at an estuary [8] Reprinted from ref. [8]. Copyright 2019, John Wiley and Sons; permission conveyed through Copyright Clearance Center, Inc.

**Figure 5 molecules-26-05469-f005:**
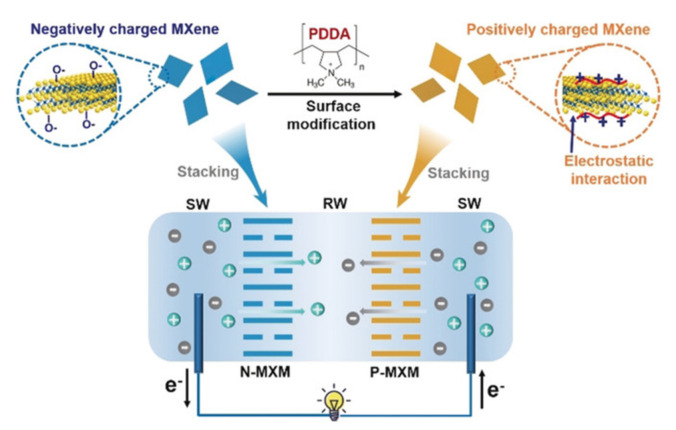
Schematic representation of harnessing osmotic energy using oppositely charged Ti_3_C_2_T_x_ MXene membranes with 2D nanofluidic ion channels. The negatively charged (N-MXM) pristine MXene and positively charged (PXM) MXene membranes can be paired to form continuous 2D nanofluidic ion channels for osmotic energy conversion. SW: seawater; RW: river water [51]. Reprinted from ref. [51]. Copyright 2020, John Wiley and Sons; permission conveyed through Copyright Clearance Center, Inc.

**Figure 6 molecules-26-05469-f006:**
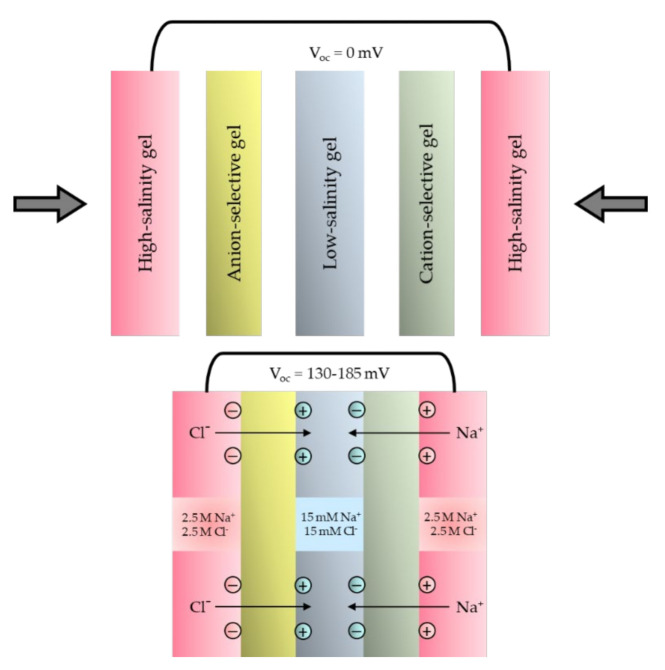
The pictorial representation of an electrochemical cell of bionic soft power sources.

**Figure 7 molecules-26-05469-f007:**
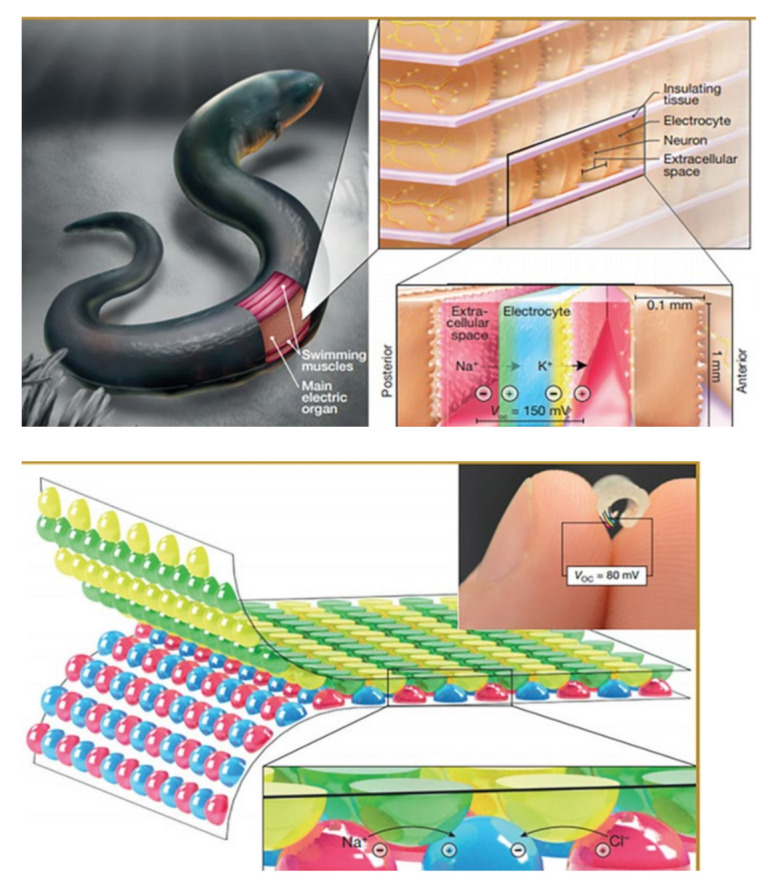
The electric organ of electric eel and bio-inspired soft power sources. Reprinted from ref. [92]. Copyright 2019, John Wiley and Sons; permission conveyed through Copyright Clearance Center, Inc. [49,92].

**Figure 8 molecules-26-05469-f008:**
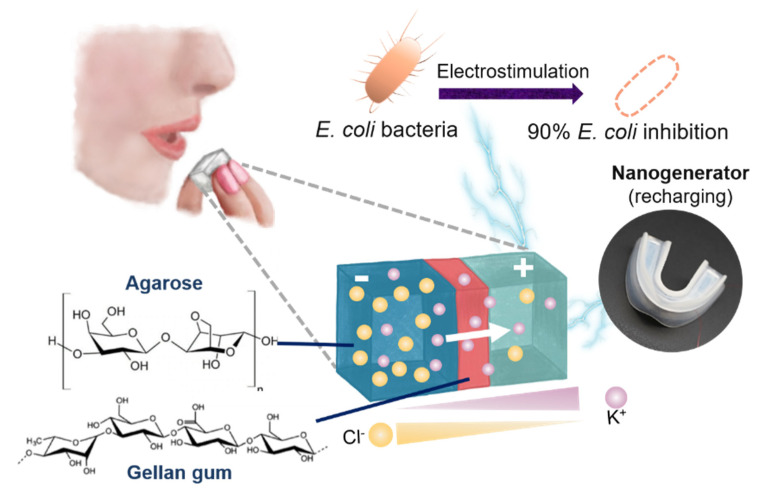
Schematic illustration of polysaccharide-based salinity gradient system for osmotic energy conversion [36]. Reprinted from ref. [36]. Copyright 2021, Elsevier; permission conveyed through Copyright Clearance Center, Inc.

**Table 1 molecules-26-05469-t001:** Different types of ion-selective membranes.

Membrane Type	Main Materials	Functional Groups/ Surfaces	Salts	Ref.
Anion selective	Ionized Wood Membrane (Positively charged aligned cellulose nanofibers)	Quaternary ammonium groups	NaCl	[8]
Anion selective	Hybrid Janus nanochannel membrane composed of two block copolymers (Poly(ethylene oxide)-block-poly(methacrylate) and polystyrene-*block*-poly(4-vinylpyridine))	Hybrid Janus nanochannel membrane	NaCl; KCl	[22]
Anion selective	3-acrylamidopropyl) trimethylammonium chloride	-NH_2_ groups	NaCl	[49]
Anion selective	Poly (sodium 4-styrene sulfonate) (PSS), hydroxypropyltrimethyl ammonium chloride chitosan (HACC)	-NH_2_ groups	Na_2_SO_4_	[50]
Anion selective	Ti_3_C_2_T*_x_* MXene membrane modified with polydiallyl dimethyl ammonium (PDDA) (positively charged)	Positively charged MXene surfaces (P-MXene)	NaCl	[51]
Anion selective	Graphene oxide membrane (GOM) (Positively charged 1-aminopropyl-3-methylimidazolium bromide (APMIB) conjugated onto GO)	Positively charged GO surfaces	NaCl	[52]
Cation selective	Ionized Wood Membrane (Negatively charged aligned cellulose nanofibers)	Carboxyl groups	NaCl	[8]
Cation selective	Ti_3_C_2_Tx MXene−Boron Nitride	-OH groups, –F groups	NaCl	[53]
Cation selective	2-acrylamido-2-methylpropane sulfonic acid	-OH groups	NaCl	[49]
Cation selective	2-hydroxyethyl methacrylate phosphate (HEMAP) hydrogel membrane	-OH groups	KCl	[54]
Cation Selective	Poly(styrenesulfonate) (PSS), anodic alumina oxide (AAO)	-OH groups, -COOH groups	KCl	[27]
Cation selective	Poly(ethylene terephthalate) (PET)	-COOH groups	KCl	[55]
Cation selective	Pristine Ti_3_C_2_T*_x_* MXene nanosheets (negatively charged)	Negatively charged MXene surfaces (p-MXene)	NaCl	[51]
Cation selective	Silk-based hybrid Membranes (composed of a silk nanofibril membrane and an anodic aluminum oxide membrane)	Negatively charged surface/ -COOH	NaCl; KCl	[56]
Cation selective	MXene/Kevlar nanofiber Composite (Ti_3_C_2_T_x_ (MXene) and charged aramid nanofiber (ANF))	(–O–) and (–OH) groups in MXene; C−N, C=O, and –COOH groups on ANF	NaCl; KCl	[57]
Cation selective	Pristine graphene oxide (GO) sheets (Negatively charged)	-COOH groups	NaCl	[52]
Cation Selective	2D kaolinite	Al-OH groups	KCl	[58]
Cation selective	Polymer/MOF hybrid membrane composed of polystyrene sulfonate (PSS)/Metal-organic frameworks (MOF) composites and anodic aluminum oxide (AAO)	Positively charged hybrid membrane surfaces	KCl	[59]
Cation selective	Ionic diode membrane	Nanochannel surfaces	KCl	[60]
Cation selective	Mesoporous Silica Thin film	Negatively charged silica surfaces	KCl	[61]
Cation selective	Nafion-filled polydimethylsiloxane (PDMS) Microchannels	Nafion	KCl	[62]
Cation selective	Nanocomposite membrane containing iron (III) oxide and sulfonated poly (2,6-dimethyl-1,4-phenylene oxide) (sPPO) polymer	Sulfonate groups (–SO_3_^-^)	NaCl	[63]
Cation selective	Polypyrrole (PPy)/chitosan (CS) composite	Amino (–NH_2_) groups	NaCl	[64]
Cation selective	Gellan gum (GG) membrane	‒COOH groups	KCl	[36]

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
