# Peer review of "Miniaturized Salinity Gradient Energy Harvesting Devices"

_molecules, 2021, doi:10.3390/molecules26185469_

Round 1
Reviewer 1 Report
The review paper is about "miniaturized salinity gradient energy harvesting devices". As described in its background, it obtains salinity gradient energy from the free energy mixture of seawater and fresh water, also known as "osmotic energy" or "blue energy", which provides a renewable and sustainable option to avoid the recent surge of global energy consumption. This research is very interesting and worth reviewing. This is the basis for the publication of this paper. I also hope that the paper will be revised so that more people can benefit from it.
1)The background of the abstract is too long and needs to be deleted.
2) The background colors of Equations 1 and 2 need to be deleted. Please check the typesetting carefully.
3) The subgraph in Figure 2 needs to be redrawn because the text is not clear.
4) In the last paragraph of the first section, what are the innovations of this review?
5) The author's comments are absent in Table 1. This is an indispensable part of the review paper.
6) There is no "Summary" in Section 2
7) There is no "Discussion" in the paper.
8) These is no "Outlook" in the paper.
Reviewer 2 Report
Dear Authors, I do not agree, that “this energy source can provide environment-friendly, …… energy production” the efforts still need to be made not only "for their optimal large-scale production" but, what is more important, for sustainable (from the environmental point of view) development of the issue.
First of all, this is not green energy. I agree we do not burn fossil fuels here, but our interference into the seaside environment is drastic. This habitat does not belong to us, authors mentioned only once about the fishes (electric fishes) only to notice their abilities to generate electric discharges.
This can be very valuable work when besides the delight about technology authors will look critically at the subject. In my opinion, this is the secret of a good, novel review article.
Author Response
We thank the reviewer for the constructive criticism as well as for providing nice suggestions.
We have revised the manuscript accordingly with the following text in the introduction section:
“Indeed, in addition to optimizing osmotic energy production, it is crucial to maintain a balanced
aquatic ecosystem. To avoid excessive utilization of aquatic resources and interference to the
living creatures in the sea environment, other naturally occurring hypersaline sources (such as
hypersaline lakes and salt domes) can be explored for obtaining greater concentration
difference and, hence, provide solutions to overcome the challenges inherent to seawater-river
water systems.”
Reviewer 3 Report
The manuscript by Hsu et al. presents a holistic review of membrane-based salinity gradient energy harvesting technology. The topic is of great interest and the manuscript reads well. Thus I recommend acceptance of the manuscript after following minor revision.
-Fig. 2. The text is barely readable. Fix the figure text.
-Diffusioosmosis: Salinity gradient also drives diffusioosmotic slip flow, which can possibly increase the energy conversion efficiency. I suggest the authors to briefly mention this effect in the main text with the following references:
Siria, A., Poncharal, P., Biance, A.L., Fulcrand, R., Blase, X., Purcell, S.T. and Bocquet, L., 2013. Giant osmotic energy conversion measured in a single transmembrane boron nitride nanotube. Nature, 494(7438), pp.455-458.
Ajdari, A. and Bocquet, L., 2006. Giant amplification of interfacially driven transport by hydrodynamic slip: Diffusio-osmosis and beyond. Physical review letters, 96(18), p.186102.
Ault, J.T., Shin, S. and Stone, H.A., 2019. Characterization of surface–solute interactions by diffusioosmosis. Soft matter, 15(7), pp.1582-1596.
Keh, H.J., 2016. Diffusiophoresis of charged particles and diffusioosmosis of electrolyte solutions. Current Opinion in Colloid & Interface Science, 24, pp.13-22.
-Line 33: Recently, Professor Tzu-En Lin and coworkers… -> Recently, Lin and coworkers…
Round 2
Reviewer 1 Report
Diccussion setion should be strength in details.
Reviewer 2 Report
Dear Authors, thank you. I expected a bit more but even this is a step in a good direction.
